# Sustainable Transformation of Waste Soft Plastics into High-Quality Flexible Sheets

**Md. Shahruk Nur-A-Tomal** [1,*] **, Farshid Pahlevani** [1] **, Saroj Bhattacharyya** [2] **, Bill Joe** [3] **, Charlotte Wesley** [1] **and Veena Sahajwalla** [1]

[1] Centre for Sustainable Materials Research and Technology (SMaRT@UNSW), School of Materials Science and Engineering, UNSW Sydney, Sydney, NSW 2052, Australia; f.pahlevani@unsw.edu.au (F.P.); c.wesley@unsw.edu.au (C.W.); veena@unsw.edu.au (V.S.)

[2] Mark Wainwright Analytical Centre, UNSW Sydney, Sydney, NSW 2052, Australia; saroj.bhattacharyya@unsw.edu.au

[3] School of Materials Science and Engineering, UNSW Sydney, Sydney, NSW 2052, Australia; w.joe@unsw.edu.au

[*] Correspondence: m.nur-a-tomal@unsw.edu.au or shahruktomal2010@yahoo.com

**Abstract:** Post-consumer soft plastics often face inadequate management practices, posing threats to both human life and the environment while also leading to the loss of valuable recyclable materials when not recycled. Traditional mechanical recycling methods are unsuitable for waste soft plastics due to their thin and flimsy nature. This study presents an effective, user-friendly process for converting waste soft plastics into new products, generating value, and expediting their collection and recycling without the need for pelletization. The outcome of this process was compared with products derived from traditional recycling and commercially available alternatives through various analytical techniques including tensile testing, Fourier transform infrared spectroscopy, differential scanning calorimetry, X-ray diffractometry, scanning electron microscopy, and energy-dispersive X-ray spectroscopy. The findings suggest that waste soft plastics can be transformed into flexible sheets without significant alterations to their properties. In particular, the ultimate tensile strength of samples recycled using the developed process in this study and traditional recycling were found to be $25.9 \pm 0.4$ and $25.2 \pm 0.8$ MPa, respectively, surpassing commercially available products by nearly 15 MPa. Additionally, a life cycle assessment revealed that producing flexible sheets from waste soft plastics using this innovative approach, rather than virgin polymer, could reduce fossil fuel depletion and global warming by 99.4% and 94.6%, respectively. This signifies the potential of the process to mitigate environmental pollution and produce high-quality products exclusively from 100% waste plastics.

**Keywords:** plastic waste; waste to valuable product; cleaner production; environmental-friendly; sustainability; circular economy

## 1. Introduction

### 1.1. The Significance of Recycling Waste Soft Plastics

Soft plastics are among the most versatile and widely utilized materials globally [1,2]. They find application in numerous sectors including packaging, household goods, agriculture, electrical and electronic devices, construction, automotive, and more. Soft plastics are prized for their unique combination of strength, ductility, barrier properties, chemical resistance, low weight, and affordability, making them an invaluable resource [3]. The magnitude of the soft plastic market becomes evident when considering that a staggering 6.9 billion plastic bags are consumed annually in Australia alone [4]. This extensive use of soft plastic products has resulted in a pressing waste management issue [5]. The proliferation of plastic products leads to an increase in post-consumer plastics [6], and it is disheartening to observe that a substantial portion of waste soft plastics is not managed

efficiently. Instead, these materials often end up in landfills, waterways, or are incinerated for energy recovery or disposal [7,8]. Plastics are generally non-biodegradable [9], with some biodegradable variants degrading very slowly and in limited quantities [10]. The persistence of waste plastics in the environment is contingent upon their chemical composition and the conditions of the disposal environment [11], ensuring their long-term presence.

A significant portion of waste plastics inevitably finds its way into water bodies including oceans, posing significant threats to marine ecosystems [12]. It is evident that many animals inadvertently ingest or become entangled in waste plastics, leading to asphyxiation, starvation, and drowning [13]. On the flip side, micro- and nano-plastics, which are degradation products of plastics, enter the human food chain through the consumption of seafood, potentially impacting human health [14–16]. Iqbal et al. [17] observed that micro- and nano-plastics could alter the soil nitrogen biogeochemistry and influence plant growth and yield. Moreover, plants absorb these micro and nano-plastics, potentially entering the human and animal food chains [17]. Microplastics have also been detected in air samples, and their inhalation or ingestion may lead to various health issues [18]. Furthermore, waste plastics can release incorporated additives or chemicals during their degradation into micro- or nano-plastics [19].

The incineration of waste plastics generates toxic emissions that are detrimental both directly through inhalation and indirectly when rain carries these toxins into water bodies and soil [20,21]. Consequently, waste plastics pose considerable risks to both life forms and the environment, thereby jeopardizing environmental sustainability [22]. Furthermore, more than 90% of polymers are derived from non-renewable petrochemicals, implying that future resources for polymer production are limited [23]. At present, 8% of the total annual fossil oil consumption is used for plastic production, with this figure projected to rise to 20% by 2050 [24]. The polymer manufacturing process also indirectly consumes petroleum resources, as approximately 3–4% of total fossil fuel consumption annually is used to power plastic production [25]. Additionally, the polymer production process emits significant greenhouse gases including carbon dioxide ($CO_2$) [26]. Climate change and fossil fuel crises have now emerged as formidable challenges to achieving sustainable development [27].

However, what is the solution to these problems? One of the most apparent solutions is to reuse already-manufactured plastics instead of producing virgin plastics from fossil resources [28]. Recycling also carries economic benefits as it reintegrates waste materials into the economy. Nonetheless, a substantial obstacle to this solution remains the lack of an economically viable process for recycling and reprocessing waste plastics.

*1.2. Development of a Recycling Method*

Only a small fraction of waste soft plastics are currently recycled due to the existing limitations of reprocessing and recycling technology. While several chemical recycling methods have been proposed to convert waste soft plastics into oil, gas, and monomers for resin production, they are not yet economically viable or environmentally sustainable [29–31]. Most waste soft plastics consist of thermoplastics, which become pliable at elevated temperatures and return to a solid state when cooled [32]. Consequently, waste soft plastics can be readily reprocessed while preserving their fundamental properties [33,34].

The traditional mechanical recycling process for waste soft plastics involves two steps, as depicted in Figure 1a: the first step entails creating pellets through processes such as pressing, shredding, extrusion, and pelletization, followed by the second step, which involves producing products through a plastic processor like cast extrusion. Nonetheless, a key challenge in recycling waste soft plastics using the traditional mechanical method lies in feeding these thin and pliable materials into plastic processing machines. Although the continuous agglomeration process has been introduced to agglomerate plastic films into manageable chunks and granulate them before extrusion and pelletization [35], this process is beset with limitations, resulting in suboptimal performance and reduced value.

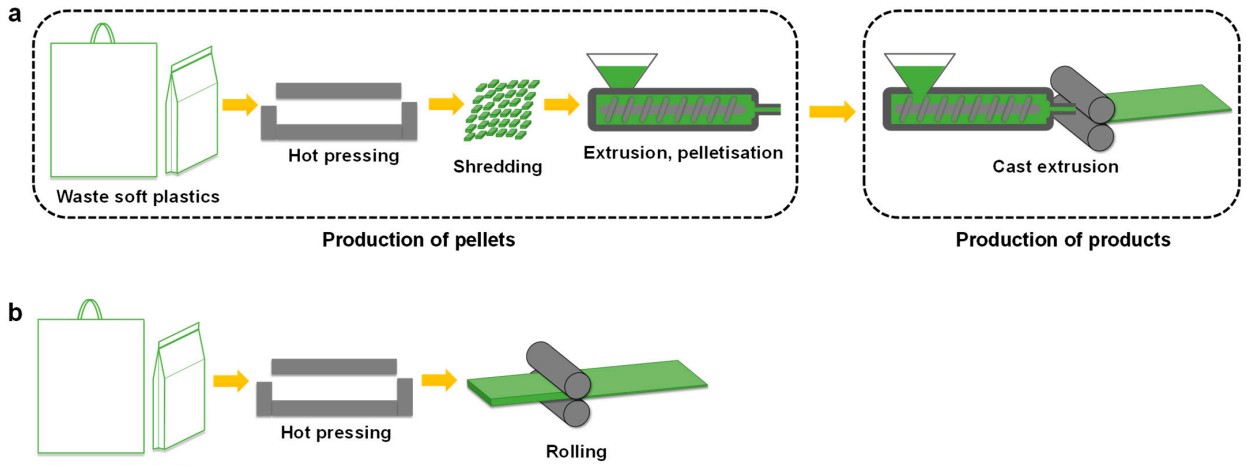

**Figure 1.** Schematic of the direct transformation of waste soft plastics by (**a**) traditional cast extrusion process and (**b**) the developed process in this study.

In this study, we propose a straightforward recycling process to directly transform waste soft plastics into high-quality flexible sheets, as illustrated in Figure 1b. This process eliminates the need for shredding and pelletization. While pelletization aids in transportation, it is more suited for rigid waste plastics, as waste soft plastics can be easily compacted into bales. Our developed process aims to maintain the properties of waste plastics by avoiding certain treatment steps that could introduce additional thermo-mechanical stress and further degrade the polymers. In addition to experimental investigations, a life cycle assessment (LCA) was conducted to quantitatively assess the environmental benefits of manufacturing plastic from waste soft plastics compared to virgin polymers.

## 2. Materials and Methods

### 2.1. Materials

In this study, we focused on recycling the most common type of post-consumer soft plastics, namely, shopping bags. This choice was made to ensure the reproducibility of our results. It is important to note that the developed process has the potential to recycle other soft plastics composed of single or compatible polymers. The collected waste soft plastics required no prior washing or cleaning, as this type of plastic is typically free of contaminants.

### 2.2. Sample Preparation

Melt Compression and Hot Pressing: The waste soft plastics were placed within a square-shaped steel die and subjected to a hot press (Carver, Wabash, IN, USA) for the purpose of melting and compressing them into a panel. The hot-press temperature was set to 160 °C, and a force of 20 MPa was applied. After the completion of melt-compression, the die was removed from the hot-press and allowed to cool in an open environment until the die's temperature reached approximately 130 °C. Subsequently, the panel was extracted from the die and directly fed into a rolling system, eliminating the need for full cooling and subsequent reheating.

Standard Cast Extrusion Process: To evaluate the influence of the processing method on product properties, a standard cast extrusion process was carried out. It should be noted that blow molding was not considered for recycling waste soft plastics due to the potential presence of contaminants that could lead to blowout of the molded product. To facilitate feeding waste soft plastics into the cast extrusion machine, granules were required, necessitating a preliminary hot-pressing and shredding step. The waste soft plastics were first hot-pressed and solidified into a dense panel. Subsequently, the panel was shredded into granules using a low-speed granulator (SHINI, UK) equipped with a 5 mm mesh. The

shredded granules were then processed into flexible sheets using a cast extrusion machine with a metering zone temperature set at 180 °C.

### 2.3. Characterization

Fourier Transform Infrared (FTIR) Spectroscopy: FTIR spectroscopy was conducted using an FTIR spectrometer (Spectrum 100, PerkinElmer, Waltham, MA, USA) equipped with an attenuated total reflectance (ATR) accessory. FTIR spectra were acquired in the range of 4000 cm$^{-1}$ to 450 cm$^{-1}$ with a resolution of 4 cm$^{-1}$. Each measurement consisted of 64 scans to reduce noise.

Differential Scanning Calorimetry (DSC) Analysis: DSC analysis was performed using a DSC (204 F1 Phoenix, NETZSCH, Selb, Germany) under an argon atmosphere, following the ASTM D3418 standard [36]. Heating and cooling were conducted at a rate of 10 °C/min, with calculations based on the second heating cycle.

X-ray Powder Diffraction: X-ray powder diffraction data were collected using MPD and Empyrean X-ray diffraction instruments equipped with a copper tube as an anode. Post-consumer soft plastic data were collected using an Empyrean system in transmission mode with incident beam optics and a focusing mirror to minimize the preferred orientation effects. The remaining samples were collected using the MPD system with programmable divergence slits in its incident optics. A PIXcel detector was used in 1-dimensional mode. To estimate crystallite size, a mixed fitting approach involving full pattern Rietveld fit and Le-Bail fit [37] was employed. Le-Bail fit was particularly useful for the polyethylene phase, given the limited available crystallographic information for this phase.

Tensile Properties: Tensile properties of the samples were assessed using a universal testing system (5982, Instron, High Wycombe, UK) with a laser extensometer (LX 500, MTS, Eden Prairie, MN, USA) in accordance with the ASTM D638 standard [38].

Microstructural Analysis: The microstructure of the samples was examined using a scanning electron microscope coupled with an energy-dispersive X-ray spectroscope (SEM/EDS, S3400 Hitachi, Tokyo, Japan). Prior to microanalysis, the samples were polished and coated with carbon to prevent overcharging.

### 2.4. Life Cycle Assessment (LCA)

To evaluate the environmental impact of the processes, a LCA was conducted following the ISO 14040:2006 standard [39]. SimaPro 9.0.0.49 software, in conjunction with the Ecoinvent 3 library and TRACI 2.1 V1.05/US 2008 method, was employed to assess the impacts of the processes. The system boundary was carefully defined and is illustrated in Figure 2. The process was divided into two stages: the polymer pellet manufacture stage and the plastic product manufacture stage. Transportation of both waste soft plastics and virgin pellets was excluded from this LCA analysis.

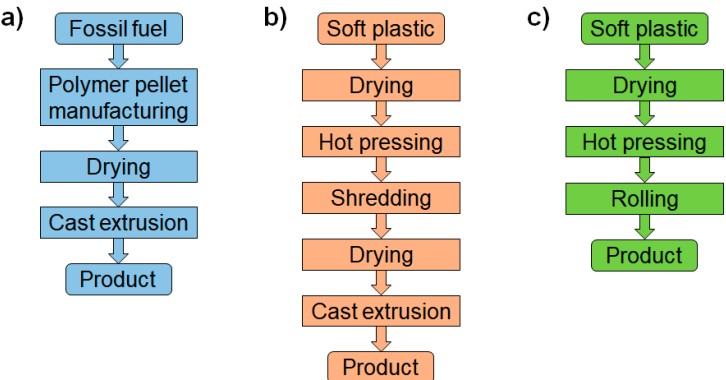

**Figure 2.** System boundaries for the (**a**) cast extrusion using virgin pellets, (**b**) cast extrusion using waste soft plastics, and (**c**) developed process in this study using waste soft plastics.

## 3. Results and Discussion

### 3.1. Effect of Remanufacturing on the Properties of Polymer

### 3.1.1. FTIR Investigation

Fourier transform infrared (FTIR) spectroscopy is an invaluable tool for the characterization of polymer materials. The FTIR spectra of the samples are presented in Figure 3. All of the spectra exhibited characteristic peaks typical of polyethylene polymers. Notably, three pairs of overlapping peaks were observed, indicating C–H symmetric and asymmetric stretching vibrations at ~2915 and ~2849 cm$^{-1}$, $CH_2$ bending vibrations at ~1472 and ~1463 cm$^{-1}$, and $CH_2$ rocking vibrations at ~730 and ~719 cm$^{-1}$ [40–42]. Additionally, peaks in the wavenumber range of 1400 to 1300 cm$^{-1}$ (as shown in Figure 3b) were used to identify the polyethylene category. According to Gulmine et al. [43], the collected soft plastic does not primarily consist of high-density polyethylene (HDPE) since it exhibits a characteristic peak at ~1377 cm$^{-1}$.

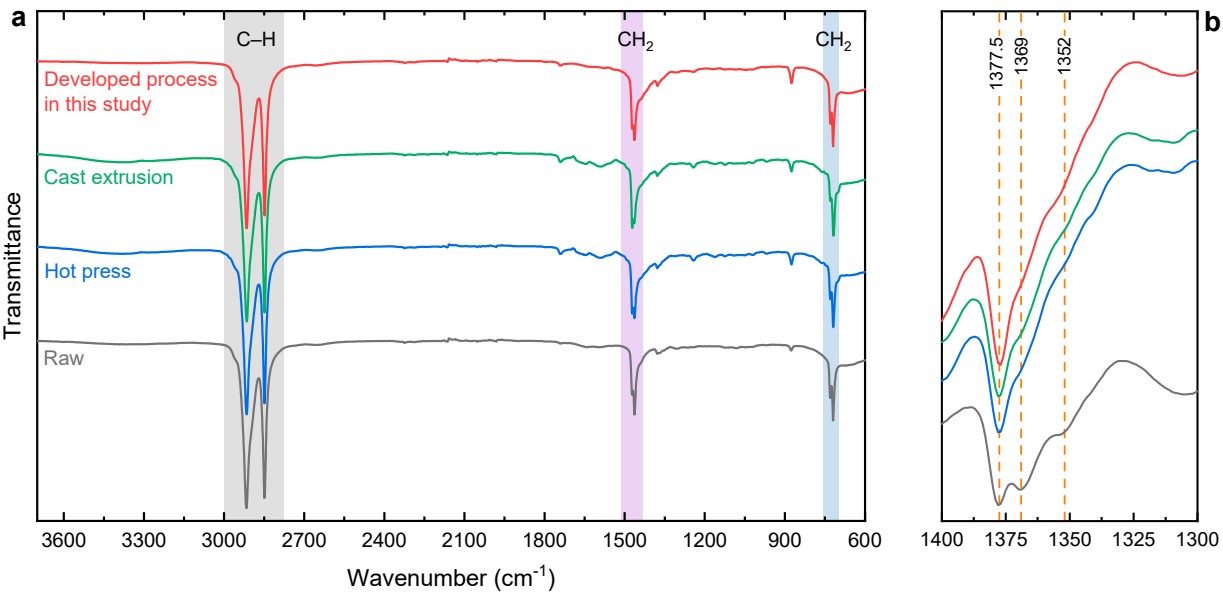

**Figure 3.** Fourier transform infrared (FTIR) spectra of the samples: (**a**) full range and (**b**) magnification of the 1400 to 1300 cm$^{-1}$ range.

Noteworthy differences were observed between the spectra of the raw and recycled samples. For post-consumer soft plastic, a small carbonyl characteristic peak at ~1740 cm$^{-1}$ was present. However, after reprocessing, both in the new process and traditional cast extrusion recycling samples, the intensity of this peak slightly increased. A minor increase was also observed in the intensity of peaks at ~1590, ~1242, 1125, and ~875 cm$^{-1}$. Conversely, the intensity of peaks at 1463 and 730 cm$^{-1}$ slightly decreased after reprocessing. Furthermore, new characteristic peaks appeared at ~1160 and 700 cm$^{-1}$ in the processed samples. These subtle changes in FTIR spectra could be attributed to potential polymer chain degradation [43].

### 3.1.2. DSC Analysis

Differential scanning calorimetry (DSC) provides insights into the material's structure. The DSC curves for the second heating of the samples are depicted in Figure 4, revealing two distinct and well-separated melting peaks. The higher-temperature peak corresponded to the melting point of linear low-density polyethylene (LLDPE), while the lower-temperature peak corresponded to the melting point of low-density polyethylene (LDPE) [44]. This suggests that the samples constitute a non-homogeneous blend of LDPE and LLDPE, resulting in phase separation [45]. It is worth noting that LLDPE is favored for its strength, while LDPE enhances processing, clarity, and gloss [46]. Blending LDPE

with LLDPE is a common practice to reduce extruder pressure and torque, improve melt strength, and enhance bubble stability [47].

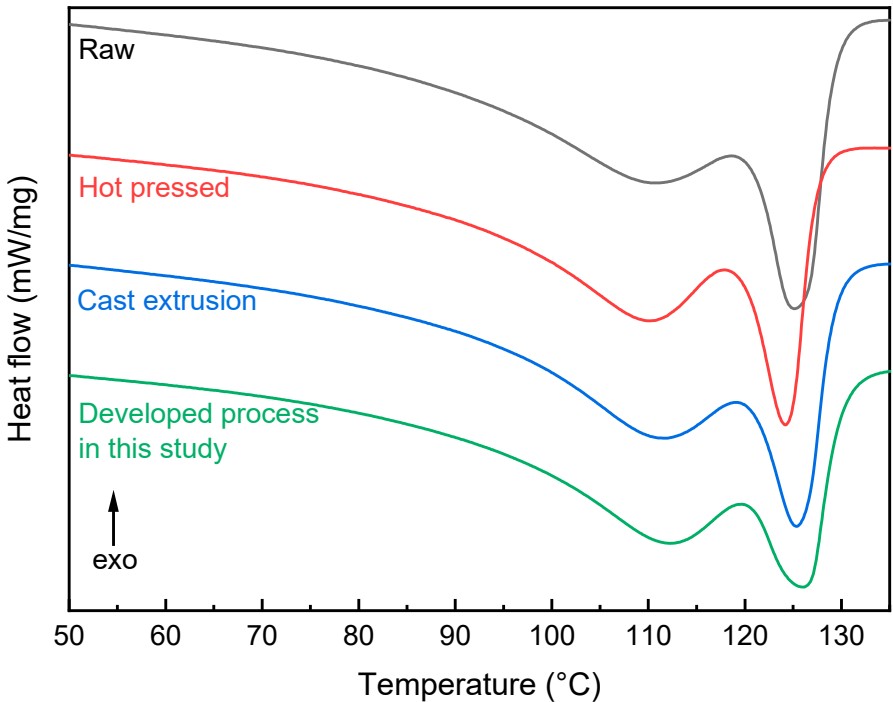

**Figure 4.** Differential scanning calorimetry (DSC) curves for the second heating of raw and recycled materials.

Figure 4 reveals that the intensity of the melting peak corresponding to LLDPE slightly decreased for both the new process in this study and the traditional cast extrusion recycling samples, while the intensity of the peak corresponding to LDPE experienced a minor increase. The peak positions and degree of crystallinity, as summarized in Table 1, are vital indicators of polymer properties. The degree of crystallinity has a direct impact on a material's properties. After reprocessing, the degree of crystallinity decreased for both the developed process in this study and the traditional cast extrusion recycling samples, approaching the original value of post-consumer soft plastic. The change in crystallinity may result from chain scission, branching, or cross-linking of polyethylene during reprocessing [48]. The degree of crystallinity of the samples was found to be higher post-recycling using the new process. A higher crystallinity in polymeric materials like polyethylene was associated with improved mechanical properties such as increased tensile strength and stiffness. It also enhanced the thermal resistance and decreased gas permeability, which is advantageous for packaging applications that require maintaining product integrity under varying environmental conditions. This increased crystallinity can affect the performance by potentially extending the material's lifecycle when used in products that are subject to mechanical stress or thermal cycling.

**Table 1.** Variation of degree of the crystallinity of raw and recycled materials.

| Sample | 1st Peak Position (°C) | 2nd Peak Position (°C) | Crystallinity (%) |
|---|---|---|---|
| Raw | 110.69 | 125.17 | 55.64 |
| Hot press | 110.17 | 124.20 | 47.80 |
| Cast extrusion | 111.68 | 125.34 | 53.41 |
| Developed process in this study | 112.27 | 126.03 | 54.35 |

### 3.1.3. XRD Study

The X-ray powder diffraction (XRD) analysis provided crucial insights into the structural composition of the recycled soft plastics. Upon examination, the XRD patterns revealed prominent peaks at approximately 20.5° and 23.8° 2θ, which are indicative of the (110) and (200) planes of semi-crystalline polyethylene, respectively as shown in Figure 5 [49]. These peaks are characteristic of the orthorhombic unit cell structure of polyethylene, signifying the presence of both low-density polyethylene (LDPE) and linear low-density polyethylene (LLDPE).

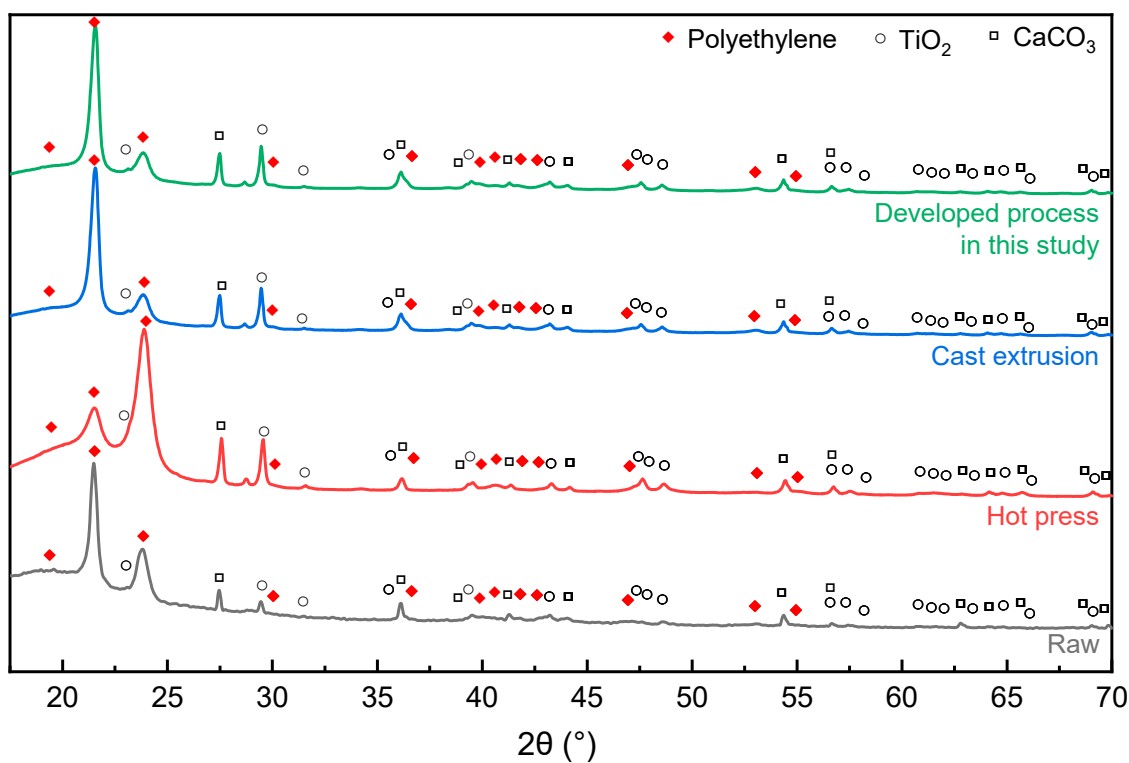

**Figure 5.** X-ray diffraction (XRD) spectra of the raw and recycled materials.

A closer inspection of the peak positions showed a slight shift toward higher angles in the recycled samples compared to the raw material, suggesting a marginal decrease in the interplanar spacing post-recycling. This observation could be indicative of a tighter packing of the polymer chains, potentially resulting from the reprocessing conditions.

The peak widths were found to broaden in the recycled samples, particularly after hot pressing, which is commonly associated with a reduction in crystallite size. This decrease in crystallite size from 46.5 nm in the raw material to 25.4 nm post-hot pressing, as presented in Figure 6, reflects the thermomechanical history imparted during recycling, likely due to the application of heat and pressure, which can disrupt the polymer chains and lead to a more amorphous structure.

Noteworthily, the subsequent processing steps appear to have partially reversed this effect, with an increase in crystallite size noted in samples produced via the new process (55.5 nm) and traditional cast extrusion recycling (57.6 nm). This suggests that the conditions during the final stage of our developed process may facilitate a certain degree of polymer chain reorganization and crystallite growth.

The crystallite size is a critical factor in determining the mechanical properties of polymeric materials. Smaller crystallites can contribute to increased material toughness by promoting energy dissipation during deformation, while larger crystallites often correlate with greater stiffness and tensile strength due to more pronounced polymer chain alignment.

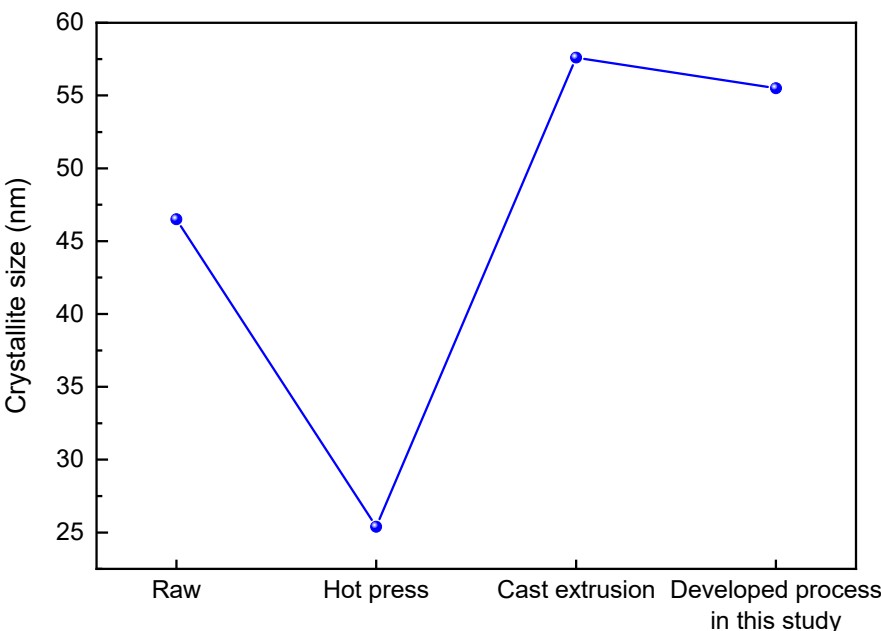

**Figure 6.** Change in the crystallite size of polyethylene during reprocessing.

The degree of crystallinity, a vital parameter influencing the mechanical properties of polymeric materials, was computed based on the relative areas under the crystalline and amorphous peaks. The analysis indicated a decrease in crystallinity immediately after hot pressing, which then approached the original value in the final recycled product. This underscores the ability of our novel process to maintain the crystalline integrity of the material, which is crucial for preserving its strength and durability.

Additives such as titanium oxide ($TiO_2$) and calcium carbonate ($CaCO_3$) were also detected, as evidenced by their respective diffraction peaks. $TiO_2$ is a well-known UV stabilizer, while $CaCO_3$ serves as a cost-effective filler that can enhance mechanical properties such as stiffness and impact strength. Their presence was uniformly maintained across the recycled samples, indicating that our recycling process did not adversely affect the dispersion of these additives.

The incorporation of these additives and their interaction with the polymer matrix play a pivotal role in the performance of the recycled plastics. $TiO_2$ is crucial for preventing UV-induced degradation, while $CaCO_3$ can contribute to the overall strength and opacity of the final product.

The XRD patterns of our recycled material exhibited a high degree of similarity with conventional recycling process, affirming the reliability of our process. The slight differences observed in peak intensities and positions offer an additional layer of validation for the effectiveness of our method in preserving the material's structural properties.

This comprehensive XRD analysis demonstrated that our developed recycling process, while introducing minimal structural alterations, preserves the integral properties of the waste soft plastics, ensuring their suitability for subsequent applications where mechanical integrity is paramount.

### 3.1.4. Mechanical Testing

Mechanical testing, specifically tensile testing, is crucial for evaluating the properties of plastic materials. Tensile testing assesses how well a product performs under tension. The test involves elongating a sample and measuring the load it carries until failure. The results are then transformed into a stress–strain curve, providing insights into mechanical properties such as ultimate strength and elastic modulus.

The tensile test results for the raw and recycled samples as well as two commercial films are illustrated in Figures 7 and 8. These results are consistent with the aforementioned

DSC and XRD findings. The elastic modulus, as seen in Figure 8, was initially low for the hot-pressed sample, but it increased for both the cast extrusion and new process samples, which corresponded to changes observed in the XRD analysis. There was a slight decrease in the ultimate tensile strength after reprocessing, likely resulting from potential chain scission and alterations in crystallinity, as indicated in Table 1 [50]. Overall, it can be concluded that the recycling process does not significantly impact the mechanical properties of the polymer. Notably, the ultimate tensile strength of the new process samples exceeded the minimum requirements for plastic films made from low-density polyethylene and linear low-density polyethylene for general use and packaging applications, as specified in ASTM D4635 [51].

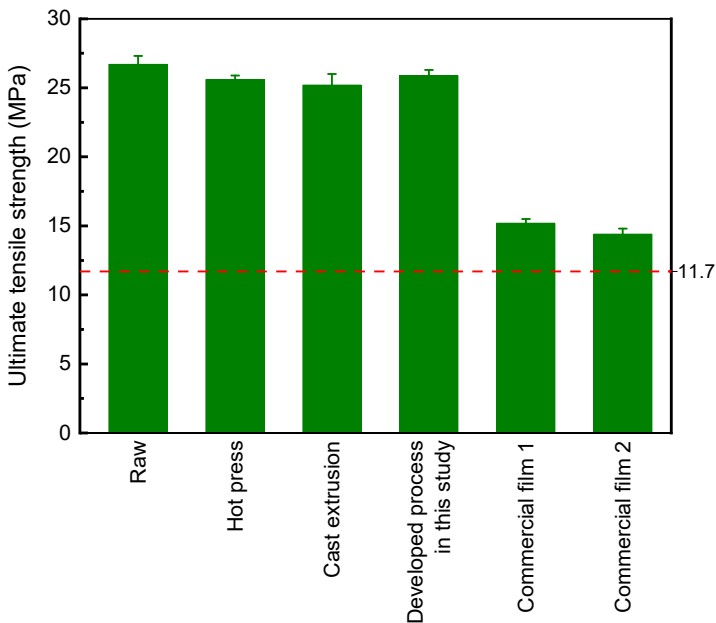

**Figure 7.** Ultimate tensile strength of the raw and recycled samples as well as two commercial products. The red dash line indicates the requirement for plastic films made from low-density polyethylene and linear low-density polyethylene for general use and packaging applications.

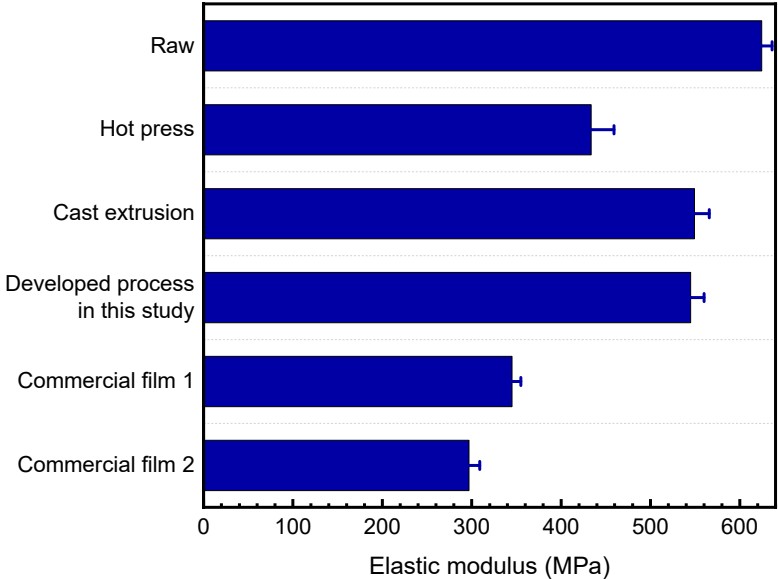

**Figure 8.** Comparative elastic modulus of the collected bag (termed as raw), different recycled material samples, and two commercially available films in megapascals (MPa).

The mechanical testing results, specifically the ultimate tensile strength and elastic modulus of the recycled materials, were contextualized to illustrate their practical implications. By comparing these values with the requirements for various commercial and industrial applications, we can evaluate the feasibility of using these recycled materials in real-world scenarios. For example, the ultimate tensile strength of our recycled material, which surpasses the industry benchmark for general-use plastic films, indicates its suitability for applications where strength is crucial. Similarly, the elastic modulus provides insights into the material's flexibility, suggesting potential uses in consumer products where both rigidity and pliability are desired. This evaluation is indispensable for manufacturers and end-users who are considering the adoption of sustainable materials as a viable alternative to virgin plastics.

### 3.1.5. Microstructural Analysis

The microstructural characteristics of the recycled soft plastic samples were thoroughly investigated using scanning electron microscopy (SEM) coupled with energy-dispersive X-ray spectroscopy (EDS). SEM images at both $100\times$ and $500\times$ magnifications, as displayed in Figure 9, provided detailed visuals of the samples' topography and particle distribution, revealing the intricate details of the materials' internal structure.

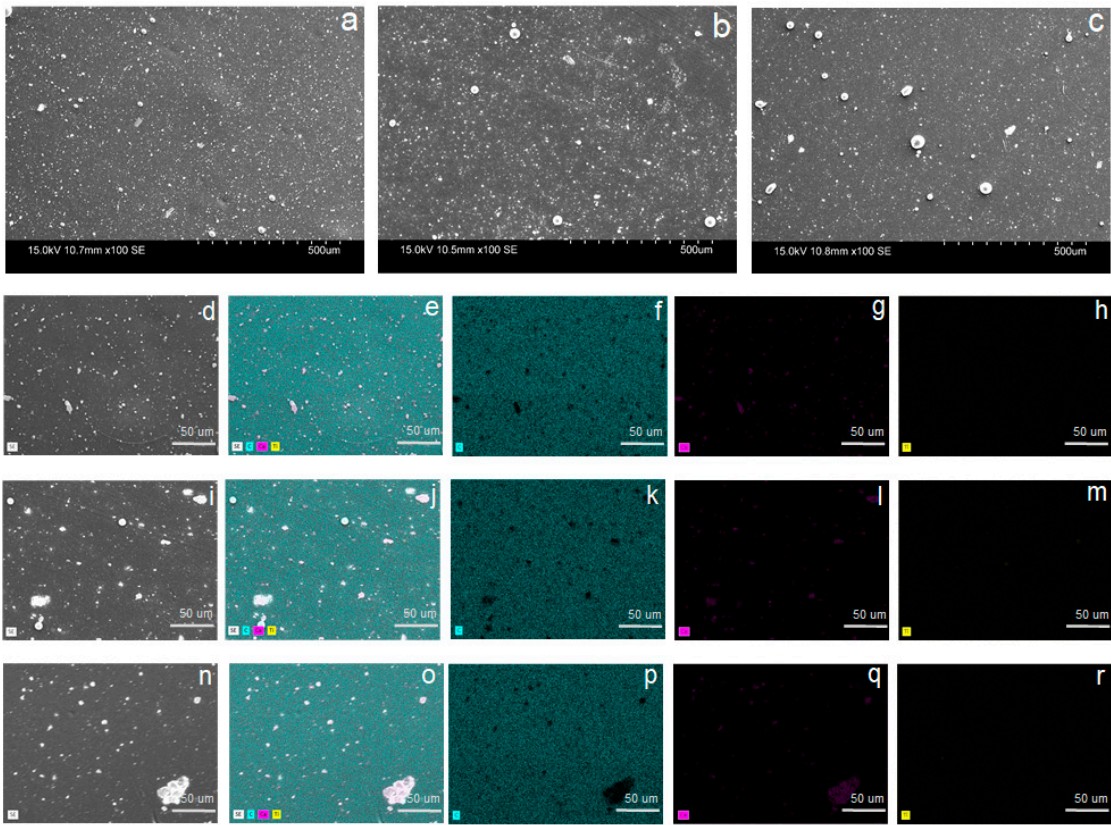

**Figure 9.** Microstructural analysis. (**a**–**c**) $100\times$ magnification SEM image of (**a**) the hot press, (**b**) cast extrusion, and (**c**) developed process samples; (**d**–**r**) $500\times$ magnification SEM/EDS mapping of the (**d**–**h**) hot press, (**i**–**m**) cast extrusion, and (**n**–**r**) developed process samples.

The SEM analysis at $100\times$ magnification for the hot-pressed, cast extrusion, and developed process samples revealed a distinct morphology indicative of the various stages of processing. The hot-pressed samples showed a denser particle distribution with smaller-sized features, suggesting a higher degree of material compaction as a result of the pressing operation. This is consistent with the concept that the application of heat and pres-

sure during hot pressing can significantly influence the microstructure, leading to the observed densification.

At 500× magnification, the SEM images provided a closer look at the dispersion of filler particles within the polymer matrix. The hot-pressed samples exhibited a fine dispersion of $CaCO_3$ and $TiO_2$ particles. The high-resolution SEM images allowed for the observation of the interfaces between the polymer matrix and the filler particles, which is crucial for understanding the interfacial adhesion and its impact on mechanical properties.

The EDS mapping complemented the SEM analysis by identifying the elemental composition associated with the various phases present within the samples. The analysis confirmed that the larger particles were predominantly composed of $CaCO_3$, while the smaller particles were identified as $TiO_2$. This distinction is vital as the size and distribution of these particles can significantly influence the mechanical strength, barrier properties, and stability of the recycled material.

The developed process samples demonstrated a uniform particle distribution with a more pronounced presence of larger particles compared to the hot-pressed samples. This could be attributed to the rolling step included in the developed process, which may facilitate a more uniform dispersion of filler particles and potentially enhance the interfacial bonding between the fillers and the polymer matrix.

To further understand the implications of these findings, it is crucial to discuss the impact of particle size and distribution on the mechanical properties of the final product. The finer particle size observed in the hot-pressed samples could be linked to an increased surface area for stress transfer, potentially leading to improved mechanical performance under certain conditions. On the other hand, the more evenly distributed and larger particles in the developed process samples may contribute to increased ductility and toughness, which are essential attributes for flexible sheet applications.

Additionally, the effects of the recycling process on the polymer–filler interface were analyzed. It was hypothesized that the shear forces and heat applied during the developed process may have promoted a better interfacial adhesion between the polymer chains and the filler particles, leading to an enhanced stress distribution throughout the material. This could explain the improved mechanical properties observed in the tensile testing results, as a robust polymer–filler interface is known to contribute to the overall strength of the composite material.

In summary, the characterization results indicate that the new process does not significantly alter the original properties of waste soft plastics. Future studies should focus on the industrial-scale application of this process. Building upon this foundation, the exploration of end-use applications for the resultant high-quality flexible plastic sheets revealed a spectrum of possibilities across various sectors. The robustness, coupled with the maintained material properties, makes these sheets particularly suitable for several non-food contact applications.

In the construction industry, the potential of these sheets for insulation and sound-proofing is supported by their inherent thermal properties. The automotive sector can also utilize these materials for non-structural interior elements, aligning with the industry's durability and safety standards. As for consumer goods, the possibilities range from furniture coverings to durable luggage, leveraging the material's aesthetic appeal.

Agricultural applications such as greenhouse tarpaulins and irrigation system components are envisioned due to the material's resilience and environmental safety. The textile industry might find value in these recycled sheets as synthetic leather, harnessing their flexibility and texture. Furthermore, the structural integrity of these sheets introduces them as a viable option for geotextile applications in civil engineering projects including soil stabilization and erosion control.

The sporting goods industry presents another avenue where the material's impact resistance and cushioning properties could be advantageous for the development of equipment and protective gear. It is imperative to underscore the importance of rigorous safety evaluations, especially since the sheets are not intended for food contact purposes due

to potential thermal treatment by-products. As part of our commitment to upholding the highest standards of health and safety, ensuring that the recycled product is devoid of hazardous substances is a fundamental aspect of this research, in strict adherence to regulatory guidelines.

Thus, the practicality of implementing the recycled sheets in these diverse applications is contingent upon their compliance with industry-specific requirements and the affirmative outcomes of exhaustive safety assessments, confirming the absence of any detrimental by-products post-recycling.

Regarding the safety of the recycled plastic sheets, particularly in relation to the thermal treatment process, we recognize the importance of ensuring that no harmful by-products compromise the integrity of the final product. While this study did not encompass a specific chemical analysis to identify potential hazardous substances like phthalates, we thoroughly reviewed the relevant literature on the thermal processing of similar plastics. Studies suggest that the processing temperatures employed in our method are below those that could result in the release of such substances. Furthermore, a study investigating the melting of recycled HDPE shopping bags, similar to our input materials, indicated that phthalate release at the applied temperatures was minimal [52]. It is worth mentioning that although small amounts of gases could be produced during the recycling steps, implementing safety precautions such as sufficient ventilation can mitigate this issue. These findings align with the regulations and standards set forth by environmental and health authorities, which guide the safe recycling of plastics. Therefore, based on the available data and regulatory frameworks, we have confidence in the safety of the recycled sheets for the proposed non-food contact applications.

### 3.2. Evaluating Environmental Impacts: LCA Results

A LCA was performed to quantitatively evaluate the environmental benefits of manufacturing plastic from waste soft plastics compared to virgin polymer. The net environmental benefit of the new process is estimated as the avoided environmental impacts of virgin polymer production minus the incurred environmental impacts of pellet preparation from waste plastic. The environmental impacts of the scenarios are detailed in Table 2. As shown in Figure 10, the new process substantially reduces the environmental impacts compared to traditional cast extrusion, even though it utilizes waste soft plastics. The LCA results underscore that the major distinction between the two processes lies in virgin pellet production.

**Table 2.** Environmental impacts of the production of 1 kg of flexible sheet through different scenarios.

| Impact Category | Unit | Cast Extrusion Using Virgin Pellets | Cast Extrusion Using Waste Soft Plastics | Developed Process in This Study Using Virgin Pellets | Developed Process in This Study Using Waste Soft Plastics |
|---|---|---|---|---|---|
| Fossil fuel depletion | MJ | 10.647 | 1.086 | 10.190 | 0.064 |
| Global warming | kg $CO_2$e | 2.905 | 2.015 | 2.439 | 0.158 |
| Ecotoxicity | CTUe | 30.671 | 25.348 | 26.967 | 2.207 |

The newly developed recycling process offers substantial environmental benefits by significantly reducing fossil fuel depletion, global warming potential, and ecotoxicity. The process conserves fossil fuels by avoiding the production of virgin polymers, which is particularly relevant in the era of dwindling fossil reserves and the imperative shift toward sustainable energy sources. This conservation is not just a matter of resource management but a step toward energy security and the adoption of circular economy principles that recognize the intrinsic value of waste materials.

In terms of climate impact, the process achieves a 94.6% reduction in the global warming potential. This dramatic decrease in greenhouse gas emissions aligns with the global commitment to mitigate climate change, helping to stabilize rising global temperatures by

curtailing the carbon footprint of plastic production. Such innovation is in lockstep with international efforts to reduce carbon emissions and transition to carbon-neutral practices.

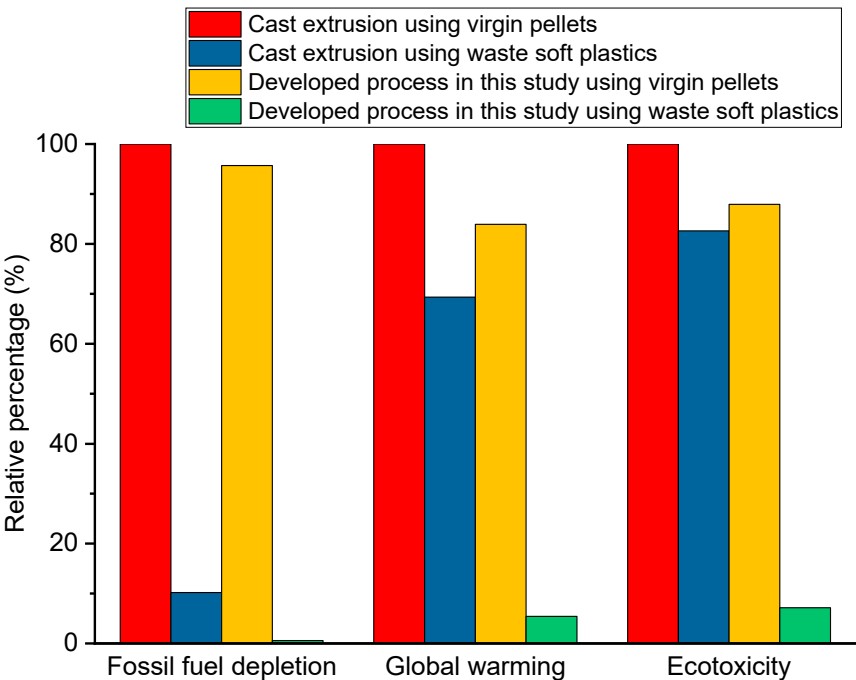

**Figure 10.** Comparison of different LCA scenarios.

Moreover, the process's 92.8% reduction in ecotoxicity marks a significant decrease in the potential to harm both terrestrial and aquatic ecosystems, ensuring the protection of biodiversity and ecosystem health. By mitigating the release of hazardous substances typically associated with plastic production, the process exemplifies an environmentally friendly approach, advancing the cause of sustainable development.

In weaving together these environmental advantages, the process emerges as a promising solution that not only mitigates pollution but also underscores a commitment to sustainable and responsible plastic manufacturing and recycling practices. It is through these lenses that the method presents a transformative step toward a greener, more sustainable future, effectively turning waste into worth, while honoring our environmental responsibilities.

It is worth mentioning that this LCA is associated solely with the processing stage of recycled plastics. This LCA does not account for the transportation and logistics phase, which represents a limitation of our current analysis. Future studies should incorporate these aspects to fully assess the environmental footprint of the recycling process. This comprehensive approach will allow for a more accurate evaluation of the overall sustainability of recycling post-consumer soft plastics.

### 3.3. Techno-Economic Analysis (TEA)

The TEA was conducted to assess the feasibility of the recycling process developed in this study when scaled up to a commercial level. The TEA considered various financial indicators including capital expenditure (CapEx) and operating expenditure (OpEx) for this evaluation.

CapEx refers to the expenses associated with non-consumable parts such as plant equipment. These costs primarily include the purchase price of the equipment, which is a significant factor in the estimation of CapEx. Conversely, OpEx encompasses the costs associated with the ongoing operations, which include the expenses for all materials, utilities (such as electricity and fuel), labor, maintenance, and insurance. The estimation of OpEx takes into account the costs of materials and utilities. For the purposes of this analysis, the recycled plastics were considered to have no cost.

The comparative data for CapEx and OpEx required to produce 1 kg of plastic sheet from both virgin and waste soft plastics are presented in Table 3. These data indicate that the process utilizing waste plastics is economically advantageous. Notably, the equipment costs for processing waste plastics are significantly lower than those for virgin plastics. Furthermore, the operational costs for the recycling process of waste soft plastics are less than those for the process using virgin plastics.

**Table 3.** Techno-economic analysis (TEA) of producing 1 kg of plastics from virgin and waste soft plastics using the newly developed process.

| Product (1 kg) from Virgin Plastic | | Product (1 kg) from Waste Plastics | |
|---|---|---|---|
| **Detail** | **Value** | **Detail** | **Value** |
| CapEx | | | |
| Equipment: (a) Dryer (b) Cast extrusion | AUD2500 AUD50,000 | Equipment: (a) Shredder (b) Dryer (c) Hot-press (d) Rolling machine | AUD5000 AUD2500 AUD4400 AUD4100 |
| OpEx | | | |
| Materials: Virgin LDPE and LLDPE Utility: Electricity (6 kWh) | AUD2.40 AUD1.98 | Materials: Waste plastic Utility: Electricity (9 kWh) | - AUD2.97 |

## 4. Conclusions

In this study, we successfully developed a straightforward process for the transformation of waste soft plastics into high-quality products. This novel approach involved the melt-compression of waste soft plastics using a hot press to create a densified panel, followed by hot rolling to produce flexible sheets. The samples obtained from this innovative process were comprehensively evaluated and compared with those from traditional recycling methods through a battery of tests including tensile testing, Fourier transform infrared (FTIR) spectroscopy, differential scanning calorimetry (DSC), X-ray powder diffraction (XRD), scanning electron microscope (SEM), and energy-dispersive X-ray spectroscopy (EDS).

The assessment results suggest that waste soft plastics can be recycled without significantly altering their inherent properties. Notably, the properties of the samples derived from the developed process in this study and those from the cast extrusion recycling process are remarkably similar. The ultimate tensile strength, elastic modulus, and other mechanical properties remain largely unaffected by the recycling process.

Moreover, the life cycle assessment (LCA) results emphasize the environmental advantages of producing flexible sheets from waste soft plastics using our unique approach compared to using virgin polymers. This process minimizes environmental impact by reducing fossil fuel depletion, global warming, and ecotoxicity, making it a more sustainable and environmentally friendly choice.

Implementation of this novel recycling process offers multiple benefits including a reduction in waste material disposed of in the environment, a decrease in the use of nonrenewable fossil fuels, and the production of valuable high-quality products entirely from waste materials. This innovative approach not only contributes to waste reduction but also advances the cause of sustainability and environmental responsibility in plastic production and recycling. Future endeavors should focus on scaling up the application of this process for industrial and commercial use, thus realizing the full potential of this sustainable and environmentally conscious solution for waste soft plastics.

**Author Contributions:** Conceptualization, M.S.N.-A.-T., F.P. and V.S.; Methodology, M.S.N.-A.-T., F.P. and V.S.; Software, M.S.N.-A.-T. and S.B.; Validation, M.S.N.-A.-T., F.P. and V.S.; Formal analysis, M.S.N.-A.-T., F.P., S.B. and B.J.; Investigation, M.S.N.-A.-T., S.B. and B.J.; Resources, V.S.; Data curation, M.S.N.-A.-T.; Writing—original draft preparation, M.S.N.-A.-T.; Writing—review and editing, M.S.N.-A.-T., F.P., S.B., B.J., C.W. and V.S.; Visualization, M.S.N.-A.-T.; Supervision, F.P. and V.S.; Project administration, V.S.; Funding acquisition, V.S. All authors have read and agreed to the published version of the manuscript.

**Funding:** This research was cooperated under the ARC Research Hub for Microrecycling of Battery and Consumer Wastes (IH190100009).

**Data Availability Statement:** The datasets generated and analyzed during the current study are not publicly available but are available from the corresponding author on reasonable request.

**Acknowledgments:** We gratefully acknowledge the technical support provided by Keith Monaghan, David Miskovic, and Irshad Mansuri. We would also like to acknowledge the facilities and the scientific and technical assistance of the Electron Microscope Unit (EMU), Spectroscopy Laboratory, and X-ray Diffraction Laboratory within the Mark Wainwright Analytical Centre (MWAC) at UNSW Sydney.

**Conflicts of Interest:** The authors declare no conflict of interest.

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
