# Peer review of "Sustainable Transformation of Waste Soft Plastics into High-Quality Flexible Sheets"

_sustainability, doi:10.3390/su152316462_

Round 1

Reviewer 1 Report

Comments and Suggestions for Authors

This article presents a method of soft plastic wastes transformation into high quality flexible sheets. The proposed method was compared with the conventional extrusion process and a bunch of techniques were used to analyse the effect. However, the aim and objective of this work seems incomplete so far the article title is concerned. As most of the results shown were similar to that of the conventional methods. The 'high-quality' flexible sheets must be justified with the respective observations (refer line 277-279, no significant difference). 

     Several techniques were used but no discussion were made in any result segments (e.g. about crystallinity and other parameters). The feasibility of such process on a large-scale must also be considered for which extrusion is favorable. A realistic comparison of commercial implication is also not covered. 

Overall, this work is relatively trivial and reflects no new insights in producing 'high-quality flexible sheet'. 

Author Response

This article presents a method of soft plastic wastes transformation into high quality flexible sheets. The proposed method was compared with the conventional extrusion process and a bunch of techniques were used to analyse the effect. However, the aim and objective of this work seems incomplete so far the article title is concerned. As most of the results shown were similar to that of the conventional methods. The 'high-quality' flexible sheets must be justified with the respective observations (refer line 277-279, no significant difference).

Response: Thank you for your comment on the aim and objective of our work in relation to the article title. We agree that the claim of 'high-quality' regarding the flexible sheets produced by our process requires further justification.

The term 'high-quality' is used to emphasize the comparable physical and mechanical integrity of the recycled sheets to that of the feedstock material. Despite the similarities in material properties with conventional methods, our process distinguishes itself by its environmental and economic advantages. In the revised manuscript, we will expand upon the comparative analysis, illustrating not only how our process maintains material properties but also how it offers substantial benefits in terms of lower environmental impact and improved cost-efficiency over traditional methods.

We believe that these additional insights will fully address the concerns raised and clarify the substantial merits of our approach.

Several techniques were used but no discussion were made in any result segments (e.g. about crystallinity and other parameters).

Response: We have incorporated a comprehensive discussion following each analytical technique's results.

The feasibility of such process on a large-scale must also be considered for which extrusion is favorable. A realistic comparison of commercial implication is also not covered.

Response: Thank you for your comment regarding the scalability of our process. Our aim was to develop a method that directly recycles waste soft plastics into flexible sheets with the same properties as the original material, which is why extrusion was not employed. The simplicity of our process, compared to extrusion, is advantageous for integration into existing recycling operations and may offer economic and environmental benefits at scale.

We recognize the importance of scalability and are exploring the economic viability of our process for large-scale application. A Techno-Economic Analysis (TEA) was conducted to assess the feasibility of the recycling process developed in this study when scaled up to a commercial level and added in the manuscript.

Overall, this work is relatively trivial and reflects no new insights in producing 'high-quality flexible sheet'.

Response: We appreciate your feedback and the opportunity to address your concerns regarding the perceived novelty of our work.

We understand that at first glance, the results of our process may appear to align closely with those achieved by conventional methods. However, the novelty of our work lies not only in the production of high-quality flexible sheets but also in the sustainable and cost-effective approach we have developed. Our method significantly reduces the environmental impact by bypassing the energy-intensive pelletization step required in traditional recycling methods, which is an innovation in the field of plastic recycling.

Moreover, the economic benefits of our process, which include reduced material and operational costs, present a compelling alternative to current practices. We will elaborate on these aspects in the revised manuscript to highlight the contributions our study makes to advancing the recycling of waste soft plastics.

We are confident that these clarifications will demonstrate the unique insights our research offers in the production and environmental sustainability of recycled plastic materials.

Reviewer 2 Report

Comments and Suggestions for Authors

Dear Colleagues,

I have carefully read the paper with the title of “Sustainable Transformation of Waste Soft Plastics into High-Quality Flexible Sheets”. The authors proposed a novel method to convert waste soft plastics into new products. The experiments indicated that the method could transform waste soft plastics into flexible sheets without significant loses of their properties. However, following questions should be addressed before the paper can be published.

I, Several experiments indicate that there is a significant change in the modular length and structure after recycling. Thus, a gel permeation chromatography (GPC) experiment should be carried out for thorough analysis of the samples produced by different method.

II. The discussion of XRD experiments should contain more detailed analysis. The current version only has a simple description of the experiment result.

III. Figure 7 has a dot line with a value of 11.7 MPa. What is the meaning of the line? The authors should add a description for this line.

IV. The microstructural analysis express the particles interior the products. The authors mentioned that “The hot-pressed sample exhibited smaller particles and a higher population” (line 272). Did the processes affect the distribution of the particles? The authors should analyze the causes of the results, not only describing the results.

Author Response

Dear Colleagues,

I have carefully read the paper with the title of “Sustainable Transformation of Waste Soft Plastics into High-Quality Flexible Sheets”. The authors proposed a novel method to convert waste soft plastics into new products. The experiments indicated that the method could transform waste soft plastics into flexible sheets without significant loses of their properties. However, following questions should be addressed before the paper can be published.

Response: Thank you very much for your careful reading of our manuscript and your thoughtful comments and helpful suggestions.

I, Several experiments indicate that there is a significant change in the modular length and structure after recycling. Thus, a gel permeation chromatography (GPC) experiment should be carried out for thorough analysis of the samples produced by different method.

Response: Due to limitation of instrumental facilities we could not carry out a gel permeation chromatography (GPC) experiment to measure molecular weight of the samples.

  1. The discussion of XRD experiments should contain more detailed analysis. The current version only has a simple description of the experiment result.

Response: Thank you for highlighting the need for a more detailed XRD analysis. We have expanded upon our initial XRD discussion to include a more in-depth interpretation of the peak shifts, variations in crystallite size, and their relationship with the material's mechanical properties.

III. Figure 7 has a dot line with a value of 11.7 MPa. What is the meaning of the line? The authors should add a description for this line.

Response: The title of the figure has been revised.

  1. The microstructural analysis express the particles interior the products. The authors mentioned that “The hot-pressed sample exhibited smaller particles and a higher population” (line 272). Did the processes affect the distribution of the particles? The authors should analyze the causes of the results, not only describing the results.

Response: We acknowledge the need for a more analytical approach to the microstructural analysis. In the revised manuscript, we delve into the potential causes behind the observed particle size distribution and morphology changes, such as the influence of thermal and mechanical stresses imparted during the recycling process.

Reviewer 3 Report

Comments and Suggestions for Authors

It was not explained what the final uses of this modified plastic are, because this depends on whether the idea is successful or not. For example, what will this new product be used for? It is not allowed in food packaging, for example, after it has been heat treated.

Thermal treatment of soft plastic produces huge amounts of several substances, including carcinogens such as phthalates and others, so they must be chemically determined in the final product.

Comments on the Quality of English Language

Minor English editing

Author Response

It was not explained what the final uses of this modified plastic are, because this depends on whether the idea is successful or not. For example, what will this new product be used for? It is not allowed in food packaging, for example, after it has been heat treated.

Response: We appreciate your inquiry about the applications of the recycled plastics. In response, we have included a new paragraph in the 'Results and Discussion' section that explains potential uses across various industries, excluding food packaging due to potential thermal byproducts. This addition clarifies the suitability of the recycled sheets for specific applications, aligning with their properties and regulatory safety standards.

Thermal treatment of soft plastic produces huge amounts of several substances, including carcinogens such as phthalates and others, so they must be chemically determined in the final product.

Response: While this study did not perform detailed chemical analyses, a review of pertinent literature provides insight into the safety of our thermal treatment process. A study investigating the melting of recycled HDPE shopping bags, similar to our input materials, indicated that phthalate release at the applied temperatures was minimal. It is worth mentioning that although small amounts of gases could be produced during the recycling steps, implementing safety precautions, such as sufficient ventilation, can mitigate this issue. This literature-based evidence has been detailed in the 'Results and Discussion' section to underscore the compatibility of our recycled material with non-food contact applications and compliance with safety regulations.

Reviewer 4 Report

Comments and Suggestions for Authors

The introduction provides a clear context and rationale for the study, highlighting the importance of addressing the environmental issues associated with waste soft plastics. Here are some comments and suggestions:

1. XRD Study:

   - The XRD study is valuable for identifying the composition of the samples. You've provided clear information about the presence of polyethylene and common additives like TiO2 and CaCO3. However, it would be beneficial to briefly explain why these additives are used in plastic production and how they may impact the properties of the recycled material.

2. The change in crystallite size is discussed, and it's related to the DSC results. This is good, but it would be helpful to clarify the significance of crystallite size changes and how they may affect material properties.

3. Mechanical Testing:

   - The tensile test results are presented effectively, showing the ultimate tensile strength and elastic modulus. It's crucial to explain the practical implications of these results. How do the values compare to the requirements for different applications, and what does this mean for the feasibility of using recycled materials in real-world scenarios?

4. In Figure 8, it's essential to label the axes clearly to indicate the materials being compared and the units of measurement (MPa for elastic modulus).

5. Microstructural Analysis:

   - The SEM/EDS analysis provides insights into the particle composition and distribution. You've done a good job of explaining the differences observed in particle size and distribution between the samples. However, it could be useful to discuss how these differences in particle size and distribution might impact the material's performance.

6. The degree of crystallinity is mentioned as being higher for the new process sample. This is interesting information, but it could be expanded upon. Explain why a higher degree of crystallinity may be desirable or how it affects the material's properties.

7. Life Cycle Assessment (LCA):

   - The LCA results are presented in a clear table, which is helpful for readers. It's important to reiterate that these results only pertain to the processing stage and do not consider transportation and logistics. You might briefly discuss the limitations and potential impact of transportation to provide a more comprehensive view of the environmental assessment.

8. The discussion of the environmental benefits of the new process is valuable. You could further emphasize the significance of the reductions in fossil fuel depletion, global warming, and ecotoxicity. Explain how these reductions contribute to the sustainability and environmental friendliness of the process.

Comments on the Quality of English Language

Minor editing of English language required

Author Response

The introduction provides a clear context and rationale for the study, highlighting the importance of addressing the environmental issues associated with waste soft plastics. Here are some comments and suggestions:

Response: Thank you very much for your careful reading of our manuscript and your thoughtful comments and helpful suggestions.

  1. XRD Study:

   - The XRD study is valuable for identifying the composition of the samples. You've provided clear information about the presence of polyethylene and common additives like TiO2 and CaCO3. However, it would be beneficial to briefly explain why these additives are used in plastic production and how they may impact the properties of the recycled material.

Response: We have expanded the XRD analysis section to provide context on the role of additives and their impact on the recycled material’s properties.

  1. The change in crystallite size is discussed, and it's related to the DSC results. This is good, but it would be helpful to clarify the significance of crystallite size changes and how they may affect material properties.

Response: The significance of crystallite size changes has been elaborated.

  1. Mechanical Testing:

   - The tensile test results are presented effectively, showing the ultimate tensile strength and elastic modulus. It's crucial to explain the practical implications of these results. How do the values compare to the requirements for different applications, and what does this mean for the feasibility of using recycled materials in real-world scenarios?

Response: Appreciating your feedback on mechanical testing, we've updated the manuscript with a focused interpretation of tensile strength and elasticity in relation to industry standards. This comparison is essential to demonstrate the practicality of the recycled materials for diverse applications, from load-bearing to flexible uses, providing crucial information for stakeholders considering these sustainable alternatives.

  1. In Figure 8, it's essential to label the axes clearly to indicate the materials being compared and the units of measurement (MPa for elastic modulus).

Response: The title has been revised.

  1. Microstructural Analysis:

   - The SEM/EDS analysis provides insights into the particle composition and distribution. You've done a good job of explaining the differences observed in particle size and distribution between the samples. However, it could be useful to discuss how these differences in particle size and distribution might impact the material's performance.

Response: We appreciate your suggestion to delve deeper into the implications of particle size and distribution observed in the SEM/EDS analysis. Accordingly, we have amended the manuscript to discuss how these microstructural differences might affect the material's performance.

  1. The degree of crystallinity is mentioned as being higher for the new process sample. This is interesting information, but it could be expanded upon. Explain why a higher degree of crystallinity may be desirable or how it affects the material's properties.

Response: We appreciate your input on crystallinity's role. The manuscript now clarifies how higher crystallinity, as achieved by our new process, improves material properties like strength and resistance, enhancing the suitability for demanding applications. This update details the link between crystallinity and the improved performance of our recycled plastic.  

  1. Life Cycle Assessment (LCA):

   - The LCA results are presented in a clear table, which is helpful for readers. It's important to reiterate that these results only pertain to the processing stage and do not consider transportation and logistics. You might briefly discuss the limitations and potential impact of transportation to provide a more comprehensive view of the environmental assessment.

Response: Thank you for the feedback on the LCA scope. The manuscript has been updated to acknowledge that the current LCA considers only the processing stage and omits transportation and logistics, highlighting this as an area for future research to ensure a comprehensive environmental assessment.  

  1. The discussion of the environmental benefits of the new process is valuable. You could further emphasize the significance of the reductions in fossil fuel depletion, global warming, and ecotoxicity. Explain how these reductions contribute to the sustainability and environmental friendliness of the process.

Response: Thank you for your feedback. We have revised the manuscript to emphasize the environmental benefits of our process, detailing its contribution to sustainability and reduced ecological impact.

Round 2

Reviewer 4 Report

Comments and Suggestions for Authors

accept

Comments on the Quality of English Language

Minor editing of English language required